# Deciphering the Interplay: Thieno[2,3-*b*]pyridine’s Impact on Glycosphingolipid Expression, Cytotoxicity, Apoptosis, and Metabolomics in Ovarian Tumor Cell Lines

**DOI:** 10.3390/ijms25136954

**Published:** 2024-06-25

**Authors:** Zdravko Odak, Sandra Marijan, Mila Radan, Lisa I. Pilkington, Monika Čikeš Botić, David Barker, Jóhannes Reynisson, Euphemia Leung, Vedrana Čikeš Čulić

**Affiliations:** 1Department of Gynecology and Obstetrics, University Hospital of Split, 21000 Split, Croatia; monika.cikes@gmail.com; 2Department of Medical Chemistry and Biochemistry, University of Split School of Medicine, 21000 Split, Croatia; sandra.marijan@mefst.hr (S.M.); vedrana.cikes.culic@mefst.hr (V.Č.Č.); 3Department of Biochemistry, Faculty of Chemistry and Technology, University of Split, 21000 Split, Croatia; mradan@ktf-split.hr; 4School of Chemical Sciences, The University of Auckland, Auckland 1010, New Zealand; lisa.pilkington@auckland.ac.nz (L.I.P.); d.barker@auckland.ac.nz (D.B.); 5Te Pūnaha Matatini, Auckland 1010, New Zealand; 6The MacDiarmid Institute for Advanced Materials and Nanotechnology, Wellington 6140, New Zealand; 7School of Pharmacy and Bioengineering, Keele University, Staffordshire ST5 5BG, UK; j.reynisson@keele.ac.uk; 8Faculty of Medical and Health Sciences, Auckland Cancer Society Research Centre, Auckland 1010, New Zealand; e.leung@auckland.ac.nz

**Keywords:** ovarian cancer, glycosphingolipids (GSLs), apoptosis, cytotoxicity, metabolomics, thieno[2,3-*b*]pyridine, cancer stem cells (CSCs)

## Abstract

Ovarian cancer is among the most prevalent causes of mortality among women. Despite improvements in diagnostic methods, non-specific symptoms and delayed gynecological exams can lead to late-stage ovarian tumor discovery. In this study, the effect of an anti-cancer compound, 3-amino-*N*-(3-chloro-2-methylphenyl)-5-oxo-5,6,7,8-tetrahydrothieno[2,3-*b*]quinoline-2-carboxamide (Compound **1**), was examined. The impacts of cytotoxicity, apoptosis, and metabolomic changes in ovarian cancer cell lines SK-OV-3 and OVCAR-3, as well as glycosphingolipid (GSL) expression, on cancer stem cells (CSCs), marked as CD49f^+^, and non-CSCs (CD49f^−^) were explored. Treatment with Compound **1** reduced the percentage of CSCs compared to non-treated cells (*p* < 0.001). The functional impact of eight GSLs on CSCs and non-CSCs was examined using flow cytometry. The glycophenotype changed in both cell lines, with increases or decreases in its expression, after the treatment. These findings raise the possibility of specifically targeting CSCs in ovarian cancer therapy. Additionally, treatment with Compound **1** resulted in statistically meaningful increased apoptosis, including both early and late apoptosis (*p* < 0.001), suggesting a pivotal role in initiating programmed cell death by the apoptotic pathway. The analysis revealed that the metabolic activity of treated cancer cells was lower compared to those of the control group (*p* < 0.001).

## 1. Introduction

Ovarian cancer is a prevalent illness, and at diagnosis, the median age of patients is 63 years old. [1,2]. In comparison, germ cell tumors are more common among individuals under 20; borderline tumors are more common in patients in their 30s and 40s, and epithelial ovarian cancer is more common in women who are 50 or older. [2]. It is the sixth most common cancer in women overall and the third most common gynecological cancer (after endometrial and cervical cancers), causing over 200,000 deaths globally per annum [3]. The strongest predictors of a decreased risk of ovarian cancer are breastfeeding, parity, tubal ligation, hysterectomy, the use of contraceptive pills, and bilateral adnexectomy [4,5,6]. Approximately 10% of patients with ovarian cancer have a genetic predisposition like *BRCA1* or *BRCA2* gene mutations or Lynch syndrome [7]. Because of this, early detection of ovarian cancer is most important for its treatment, as auspicious treatment outcomes are better within the early stages of the disease [3]. Regrettably, most ovarian tumors are detected in the later stages of development, mostly due to the non-specificity of symptoms and overdue gynecological examination [8]. Tests evaluated for screening generally fail to diagnose ovarian cancer promptly enough to decrease the death rate, and they have led to unnecessary surgical procedures for false-positive results [9].

Cancer stem cells (CSCs) are a rapidly expanding subset of tumor cells that can replicate and reappear as primary tumors [10]. Because of their ability for self-healing and initial tumor recurrence, CSCs are possible targets for therapeutic efforts. [10]. CSCs in ovarian cancer are usually defined by CD44^+^, CD117^+^, CD24^+^, CD133^+^, CD49f^+^, or the Aldehyde dehydrogenase (ALDH)^+^ phenotype. They are considered to be the cause of treatment resistance in several malignancies, including ovarian cancer [11].

Another possible biomarker for ovarian cancer is glycosphingolipids (GSLs), which are crucial segments of the cell plasma membrane and are composed of hydrophilic carbohydrate residues and hydrophobic ceramides. Many cellular activities are regulated by them, including adhesion, proliferation, apoptosis, recognition, alterations in signaling channels, and metastasis [12,13].

A complex disease requires complex forms of treatment, and in the case of ovarian cancer, there are not many options. For this reason, it is necessary to find new ways of treating ovarian tumors, especially those that are diagnosed late in the development of the disease [7,8], and newly synthesized compounds could fulfill this role.

Thieno[2,3-*b*]pyridines were first discovered using virtual high-throughput screens (vHTS) to find possible modifiers of phospholipase C isoforms. They were subsequently shown to have antitumor effects on numerous tumor cell lines, including ovarian tumor cells [14]. Thieno[2,3-*b*]pyridine derivatives are known to moderate multiple biological targets, such as G protein-coupled receptor (GPCR); P2Y12 platelet receptor; a DNA repair enzyme; tyrosyl DNA phosphodiesterase 1; colchicine binding site tubulin; phospholipase C-δ1; PIM1-like kinases; and eEF2K, elongation kinases eukaryotic factor 2, and cyclooxygenase [15]. Thieno[2,3-*b*]pyridine derivatives have also been shown to change the expression of glycosphingolipids (GSLs) in the cellular plasma membrane of many tumor CSCs and non-CSCs [16].

This study aimed to elucidate the potential effect of thieno[2,3-*b*]pyridine derivatives on ovarian cancer cells and obtain insight into possible mechanisms of action, which may allow for the potential development of these compounds into a new drug and, therefore, help in treating this cancer. We started our research with 4 different thieno[2,3-*b*]pyridine. 3-Amino-*N*-(3-chloro-2-methylphenyl)-5-oxo-5,6,7,8-tetrahydrothieno[2,3-*b*]quinoline-2-carboxamide (Compound **1**) was found to be the most potent, and its structure and mechanisms of action on other cancer types are known [15,17] (Figure 1).

## 2. Results

### 2.1. Cytotoxicity of Compound ***1***

Initially, the cell survival of various cell lines was examined, at different times of exposure, using the 3-(4,5-dimethylthiazol-2-yl)-2,5-diphenyltetrazolium bromide (MTT) test. Against the OVCAR-3 cell line, Compound **1** showed significant cytotoxicity at 50 nM after 48 h (75% of the cells survived), whilst 50% of metabolically active cells remained after 48 h at a concentration of 5 µM. The highest noted cytotoxicity was also at 5 µM, and after 72 h treatment only 45% of cells survived (Figure 2a).

Regarding the SK-OV-3 cell line, treatment with Compound **1** at a concentration of 50 nM showed cytotoxicity after 48 h, with 85% of cells remaining metabolically active. As expected, the fewest metabolically active cells were measured after 72 h treatment with 10 µM Compound **1**, which left only 45% of cells active (Figure 2b).

The results in Figure 2 demonstrate the time-dependent reliance between concentration and cytotoxicity. Interestingly, in both cell lines, the lowest percentage of metabolically active cells was after 72 h of exposure at a concentration of 5 µM, not with the higher 10 µM dosing for the same treatment period.

Finally, the IC_50_ for the SK-OV-3 cell line after 48 h of treatment with Compound **1** was 5.5 µM, and for the OVCAR 3 cell line, it was 5.0 µM.

### 2.2. Programmed Cell Death—Apoptosis

To test the effect of compound treatment on apoptosis, both cell lines were treated with 5 µM of Compound **1** for 48 h. It was determined that there was a significant increase in the ratio in early, late, and overall apoptotic cells after treatment compared to non-treated cells (Figure 3).

In the OVCAR-3 cell line, treatment with Compound **1** caused an increase in the percentage of cells in early apoptosis (1.29 ± 0.61% without and 5.37 ± 0.56% with treatment, *p* < 0.01), late apoptosis (1.03 ± 0.76% without and 7.07 ± 1.2% with treatment, *p* < 0.01), and overall apoptosis (2.32 ± 1.35% without and 12.64 ± 1.84% with treatment, *p* < 0.01).

In the SK-OV-3 cell line, the percentage of cells in early apoptosis was 5.45 ± 1.6% after treatment and 0.69 ± 0.11% in the control group (*p* < 0.01). In late apoptosis, after treatment, there was 1.22 ± 0.28% and 0.41 ± 0.21% in the control group (*p* < 0.05), and in overall apoptosis, there was 6.67 ± 1.87% in the group of compound-treated cells and 1.34 ± 0.48% in the group of non-treated cells (*p* < 0.01).

The representative dot blot graphs show that in comparison to the untreated cells, the treated cells display an increase in early cell apoptosis (Annexin V^+^/PI^−^ subpopulation) in both cell lines (Figure 4a,b). These results show an increase in both early and late apoptosis, suggesting that Compound **1** has a strong effect on the induction of cell death through the apoptotic pathway.

### 2.3. Cancer Stem Cells (CSCs)

We then treated both cancer cell lines with 5 µM of Compound **1** and, after 48 h, calculated the percentage of CSCs, defined as CD49f^+^. We observed that both cell lines had similar results, although the OVCAR-3 cell line had a significantly smaller subpopulation of CSCs. OVCAR-3 is, in general, less invasive than the SK-OV-3 cell line [13]: treatment with Compound **1** significantly reduced the percentage of CSCs in the SK-OV-3 cell line, compared to the non-treated cells, with an almost 40% reduction (26.47 ± 1.41% and 45.21 ± 2.41%, respectively, *p* < 0.001, Figure 5a).

The percentage of CSCs in the OVCAR-3 cell line was significantly reduced, around 30%, after treatment (0.51 ± 0.08%) compared to the non-treated cells (0.72 ± 0.13%, *p* < 0.05, Figure 5b).

### 2.4. Expression of Glycosphingolipids on Stem and Non-Stem Ovarian Cancer Cells

To determine if the different content of membrane glycosphingolipids (GSLs) was related to the cytotoxic mechanism of Compound **1**, the expression of GSLs on cell subpopulations defined as CD49f^+^ or CD49f^−^ in both cell lines, OVCAR-3 and SK-OV-3, was evaluated.

All cells, both CSCs and non-CSCs, in the SK-OV-3 cell line showed a changed glycophenotype after treatment with Compound **1**. The percentage of CSCs (CD49f^+^) positive for GSL decreased with statistical significance for seven out of the eight observed GSLs: GD3, GM2, GM3, IV^3^Neu5Ac-nLc_4_Cer, GalNAc-GM1b, Gg_3_Cer, and Gb_4_Cer, respectively (*p*-value < 0.05 in all cases). The percentage of CSCs positive for nLc_4_Cer also decreased after treatment, but this result was without statistical significance (Figure 6a). The geometric mean value of fluorescence intensity (GMI) also showed a statistically significant change in only two marked GSLs (nLc_4_Cer and GM2). Interestingly, the GMI of nLc_4_Cer was increased after the treatment, but the GMI of GM2 was seen to decrease (Figure 6c).

Furthermore, in the non-CSC population (CD49f^−^), there was an increase in positive non-CSCs marked with each of the eight observed GSLs, with statistical significance in six out of eight cases (Figure 6b). The GMI of non-CSCs (CD49f^−^) decreased in all eight observed GSLs, but only GD3 (*p* < 0.05) had statistical significance. (Figure 6d).

Figure 7a shows that the percentages of GSL^+^CD49f^+^ cells of the OVCAR-3 cell line also changed with treatment with Compound **1**, but none were statistically significant. The percentage of GSL^+^CD49f^−^ cells of the OVCAR-3 cell line was increased for all eight observed GSLs, with statistical significance being found in four GSLs, in cells with or without treatment (Figure 7c). The GMI of the CSCs (CD49f^+^) had no statistical significance for all the eight GSLs in the treated or non-treated cells, while the CD49f^−^ cells had increased GMI values for all eight GSLs, with a statistically significant increase in five of the GSLs (GD3, GM2, IV^3^Neu5Ac-nLc_4_Cer, GalNAc-GM1b, and Gg_3_Cer), as shown in Figure 7b,d.

### 2.5. Metabolomics

We also compared the metabolic response of SK-OV-3 and OVCAR-3 cell lines with and without compound treatment. Using GC-MS for metabolic profiling, 20 metabolites were found in the OVCAR-3 cell line, and 21 were found in the SK-OV-3 cell line (Table 1). Only substances mentioned in the Human Metabolome Database (HMDB4.0) were selected. The results observed for OVCAR-3 are more significant than those observed for SK-OV-3, according to the performed Student’s *t*-test. Table 1 shows that in the OVCAR-3 treated cells, eight metabolites were significantly different (*p* < 0.05) in comparison to the non-treated cells, whilst in the SK-OV-3 cell line, only two were different among the treated and non-treated cells.

We then investigated the metabolic differences in both treated cancer cell lines and conducted a comparison to their control (untreated) counterparts using principal component analysis (PCA). The PCA score plots for the OVCAR-3 cells show how the treated groups cluster differently from the control groups (Figure 8a), and in the figure below, it one can clearly distinguish between the control group and the group treated with Compound **1**. The first principal component (PC1), located on the *x*-axis and explaining 67.6% of the variability, contains most of the information and variability in the original data. The second principal component (PC2), located on the *y*-axis and explaining 28.2% of the variability, still contains a significant amount of information about the structure of the data. The high level of variation in the original data (~96%) indicates that the score plots shown in Figure 8a are an excellent representation of the variation in the data. As for the SK-OV-3 cell line, we can see that PC1’s value is 48.6%, while PC2’s is 34.1%, and the direction of the metabolites of the control and after the effect of Compound **1** is less apparent, with notable overlap in the treated and untreated groups (Figure 8b). This analysis, supporting the earlier findings, highlights the greater effect of Compound **1** on OVCAR-3 cells compared to SK-OV-3 cells.

The analysis indicates that these compounds are present in differing levels after exposure to Compound **1**, compared to the control group. The use of quantitative enrichment analysis (EA) enabled the identification of concentration patterns of metabolites and provided insight into potential biological mechanisms. The ranking of molecular pathways was based on the *p*-value found in the compound list of each pathway for a specific number of significantly changed metabolites. The treated OVCAR-3 cancer cell line had a major impact (*p* < 0.001) on the glucose–alanine cycle, galactose, sphingolipid, nucleotide sugar metabolism, and lactose degradation (Figure 9a). It is interesting that in the case of the treated SK-OV-3 cancer cells, the glucose–alanine cycle, nucleotide sugar metabolism, and steroidogenesis are the most represented factors. Still, none of those changes were statistically significant (*p* > 0.06 and higher), as shown in Figure 9b.

Using a correlation matrix and heat map, we showed an even more notable connection between metabolites after treatment with Compound **1**. Adding the “clustering” outside of the correlation matrix, we grouped certain substances. By doing so, we gained further information regarding the correlations, positive or negative, within the groups of metabolites (Figure 10a,b). It can be seen that for the OVCAR-3 cells in particular, there are groups of highly positively correlated metabolites.

### 2.6. Toxicological Profile of the Thieno[2,3-b]pyridines

The selective toxicity of the thieno[2,3-*b*]pyridines class of compounds was previously tested in various cell lines provided by the National Cancer Institute (NCI60 panel), including seven ovarian cancer cell lines, and these data were published in one of our early papers [18,19].

Furthermore, two thieno[2,3-*b*]pyridines derivatives (**6** and **7**, see Figure 11) were selected for the mouse toxicity assay in the Drug Therapeutic Programme, the National Cancer Institute [20]. Three female athymic nudes were dosed for 20 days with 100, 200, and 400 mg/Kg/dose intraperitoneally for each compound. All the mice survived the regime, suggesting that the compounds are safe or tolerated at these high doses (see the data in the Appendix A, specifically in Appendix A, not published previously).

## 3. Discussion

To the best of our knowledge, 3-amino-*N*-(3-chloro-2-methylphenyl)-5-oxo-5,6,7,8-tetrahydrothieno[2,3-*b*]quinoline-2–carboxamide (Compound **1**) has not been investigated thoroughly on ovarian tumor cells. It has only been tested for its selective toxicity on the various cell lines in the NCI60 panel, which includes, among other tumor cell lines, seven ovarian tumor cell lines [18]. For breast tumor cells, Marijan et al. found that the maximum cytotoxicity of Compound **1** occurs after 72 h at a concentration of 5 µM [17]. In our research, the IC_50_ after 48 h was 5.5 µM for SK-OV-3 cells and 5.0 µM for OVCAR-3 cells, showing Compound **1** to be more cytotoxic to ovarian cancer cells than to breast cancer cells. Furthermore, the mouse toxicity assay showed that Compound **1** was safe and tolerated at high doses in athymic nude mice.

It is especially intriguing to observe the form of cell death that Compound **1** induces in ovarian cell lines. It has been shown that primarily programmed cell death (apoptosis) takes part in this cell turnover process rather than non-programmed cell death (necrosis). In both cell lines, the treated cells displayed a substantial rise in apoptotic cell percentage. In the SK-OV-3 cell line, we observed a more prominent effect on early apoptosis, with a greater than seven times higher percentage in the treated cells compared to the control cells, while the rate of cells in late apoptosis was around twice that in the treated cells compared to the non-treated cells. In the OVCAR-3 cell line, there was an approximately four times higher percentage of cells in early apoptosis compared to the non-treated ones, while the percentage was almost seven times higher for treated cells in late apoptosis.

It has been demonstrated that various signaling pathways control CSCs’ ability to differentiate, maintain themselves, and resist drugs. Finding the signaling mechanisms that control CSCs is essential for eliminating them, which will then help leverage medication resistance and tumor recurrence [20]. As a result, the substantial decrease in the proportion of CSCs following treatment with Compound **1**, in contrast to the untreated cells, is profoundly significant. While most conducted research so far has been based on common markers of CSCs, such as CD133, CD44, or ALDH, we investigated the effect on CSCs marked as CD49f^+^. CD49f^+^, also known in the literature as integrin α-6 (ITGA6), is poorly studied in relation to ovarian cancers, but it is known that it is overexpressed in SK-OV-3 cisplatin-resistant cells [21]. The SK-OV-3 and OVCAR-3 cell lines are both derived from ovarian tumors, and both lines originate from the ascites of a woman who suffered from ovarian adenocarcinoma. OVCAR-3 is considered a high-grade serous carcinoma, whilst SK-OV-3 is a non-serous carcinoma [13]. OVCAR-3 has a much lower percentage of CSCs compared to SKOV-3 and is more sensitive to the cytotoxic effect of Compound **1** under the same treatment conditions. The percentages for CD49f^+^ (or ITGA6) were about 30% lower in the SK-OV-3 cell line and 40% lower in the OVCAR-3 cell line after treatment with Compound **1**. This can be explained by Compound **1**’s specific targeting of the membrane receptors in cancer cells and disruption of signaling pathways important for cell functioning.

The range of changes observed in GSL expression following treatment with Compound **1** reflects the diverse outcomes documented in the existing literature. The monosialylated GSLs GM2 and GM3 reduce the malignancy of tumor cells. Hakomori et al. showed in their research that GM3 inhibits the activation of growth factor receptors (GFRs), especially the epidermal growth factor receptor (EGFR) [22]. On the other hand, Huang et al. showed that overexpression of GM3 can reduce apoptosis and drug resistance in the SK-OV-3 cell line **[23]**. In our research, the percentage of GM3 expression on the SK-OV-3 cell line was decreased, whilst in OVCAR-3, it was elevated, which means further research must be conducted to explain these differences in the two cell lines.

Furthermore, after Compound **1** treatment, the percentage of cells that express ganglioside nLc_4_Cer or its sialylated version, IV^3^Neu5Ac-nLc_4_Cer, in the SK-OV-3 cell line was decreased, but expression increased in the OVCAR-3 cell line. This change is more expressive on non-CSCs of both cell lines compared to CSCs. Sialylated glycosphingolipids present on cell membranes can mediate the processes of metastasis. Sung et al. showed that the inhibition of interactions between sialylated glycolipids and their receptors on metastatic cells could prevent or slow metastasis [24].

Changes es in the expression of GSLs such as GM2, GM3, nLc_4_Cer, and IV^3^Neu5Ac-nLc_4_Cer on CSCs after treatment with Compound **1** may indicate the downregulation of the ABCG2 (ATP-binding cassette sub-family G member 2) transporter. ABCG2 can affect the level and distribution of GSLs in cells by regulating their uptake, secretion, or recycling. For example, ABCG2 has been identified as a key factor in the regulation of ganglioside levels in cells, and mutations in *ABCG2* genes are associated with changes in membrane lipid composition and ABCG2 transporter function. The ABG2 transporter is already a target in the treatment of cancer [25,26].

The percentage of cells that express different neutral GSLs with terminal GalNAc (*N*-Acetylgalactosamine) varied for both cell lines. Namely, the Gb_4_Cer expression noticeably declined after Compound **1** treatment in CSCs of both cell lines, while the percentage of Gb_4_Cer^+^ non-CSCs was increased after treatment. The changes were more prominent in the SK-OV-3 cell line, where the changes were statistically significant. The research of Tanaka et al. demonstrated that ovarian cancer cells with an elevated expression of Gb_4_Cer showed greater resistance to chemotherapeutics [27]. Comparing our results with theirs, we once again demonstrated the antitumor activity of Compound **1** and its potential use as an antitumor drug, as the expression of Gb_4_Cer on CSCs was decreased after treatment with Compound **1**.

We also observed changes in Gg_3_Cer expression after treatment with Compound **1**. On the CSCs and non-CSCs of the OVCAR-3 cell line, both the percentage and GMI were increased. The expression of Gg_3_Cer was decreased in CSCs of the SK-OV-3 cell line and increased in non-CSCs, which is in line with the research Marijan et al. conducted on breast CSCs. They explained the reduction by the deletion of lactosylceramide 4-alpha-galactosyltransferase (A4GALT), which is an essential enzyme in the epithelial-to-mesenchymal transition and can increase chemoresistance [17,28].

The percentage of treated cells positive for GalNAc-GM1b was increased in both ovarian cancer cell lines on non-CSCs and decreased in CSCs, and this result has not previously been described in the literature. A decreased percentage of CSCs that express GalNAc-GM1b may indicate a slowing down of glycolysis in CSCs, which further affects the differentiation of CSC and non-SCC phenotypes [17,26].

The expression of GD3 was increased in the non-CSCs of both cell lines after compound treatment, as it was in the CSCs of the SK-OV-3 cell line. Conversely, the percentage of GD3^+^ CSCs in the OVCAR-3 cell line decreased. GD3 is a GSL specific to tumorous cells and is absent in normal cells, causing immune system suppression and allowing tumors to become immunity-evading [29]. OVCAR-3 cell lines have a significantly lower percentage of CSCs, and this may be an explanation for the results obtained.

In our research, we have shown that Compound **1** had a diverse effect on the expression of metabolites after treatment on both cell lines, changing the expression of metabolites by increasing or decreasing their concentrations, altering metabolic pathways, and affecting cell signaling.

Interestingly, the concentration of inositol was significantly reduced in the OVCAR-3 cell line, while it was statistically significantly increased in the SK-OV-3 cell line. Considering that both cell lines are ovarian tumor cells, the differences in the metabolic response to compound 1 could potentially be linked to inherent differences in the biological characteristics of these cells.

Searching the literature, we found no previous research that could be specifically compared to our metabolomics results. However, Alarcon-Zapata et al. pointed out the diversity of behavior of SK-OV-3 and OVCAR-3 ovarian tumor cell lines in their research, where they demonstrated that the SK-OV-3 and OVCAR-3 cell lines display different features [30]. In comparison to each other, SK-OV-3 cells demonstrate accelerated migration, increased invasiveness, and more extensive metastases in vivo models. Both cell lines develop resistance to chemotherapeutic drugs such as cisplatin, although SK-OV-3 exhibits a higher rate of treatment survival [30]. This can explain why our results displayed different metabolic expressions in each cell line after treatment with Compound **1**. Pervan et al., in 2022, also confirmed changes in the metabolic pathways of breast tumor cells after treatment with cytotoxic thieno[*2,3*-b]pyridines, and the concentration of various metabolites after treatment further confirmed the importance of Compound **1** in its antitumor effect [26].

Significantly, we noted that inositol was reduced after treatment with Compound **1**. Compound **1** acts on the isoform of phospholipase C (PLC), which participates in the metabolism of inositol in cells. An increased need for inositol may be present in cancer cells due to an increased need for membrane lipids and metabolic reprogramming that supports rapid tumor growth [31]. Furthermore, the increased level of inositol can contribute to the proliferation and survival of cancer cells, which makes it a potential target for cancer therapy, and we have proven this by observing the effect of Compound **1** on them [32]. Searching the literature, we did not find similar results on ovarian tumor cells.

Also, very significantly, the concentration of heptanoate was reduced after treatment with Compound **1**. Since inositol is a precursor of heptanoate in some metabolic pathways, changes in inositol metabolism caused by PLC inhibition may affect the availability of heptanoate, which further affects its reduced synthesis. The role of heptanoate can be complex and can have different implications for the proliferation, survival, and metastasis of cancer cells, which also makes it a potential target in antitumor therapy [33].

Cancer cells usually display a modified metabolic pathway referred to as the “Warburg effect,” wherein they prioritize anaerobic glycolysis to produce energy, even when there is an environment with sufficient oxygen [34,35]. Interestingly, glucose is statistically elevated in ovarian cancer cells of the SK-OV-3 line (*p* < 0.05), while in the OVCAR-3 cell line, the level of glucose is statistically significantly decreased after treatment with Compound **1** (*p* < 0.001). Although it is to be expected that the level of glucose will be reduced in the medium around tumor cells due to the increased consumption of glucose, there may be an accumulation of glucose inside tumor cells, like in the SK-OV-3 cell line, due to altered metabolism and disturbances in regulation; however, we obtained conflicting results.

## 4. Conclusions

Our study reveals several important observations on the potential of using thieno[2,3-*b*]pyridine compounds, particularly the derivative 3-amino-*N*-(3-chloro-2-methylphenyl)-5-oxo-5,6,7,8-tetrahydrothiene[2,3-*b*] quinoline-2-carboxamide (Compound **1**), in the treatment of cancer cells.

First, our study showed that Compound **1** had significant cytotoxic effects on ovarian cancer cell lines SK-OV-3 and OVCAR-3, surpassing the results observed in previous studies in breast and prostate cancer cells. This high cytotoxicity, coupled with a preference for inducing apoptosis over unprogrammed cell death pathways, highlights its potential as a promising antitumor drug.

The effect of Compound **1** on CSCs labeled as CD49f^+^ showed that their percentage changed significantly after treatment. If we consider the role of CSCs in the cycle of tumor regrowth and the development of resistance to drugs, this could make cancer CSCs a potential target.

In addition, our study investigated in detail the changes in GSLs and metabolites after treatment. Notably, the changes in GSL expression, especially the downregulation of some gangliosides associated with metastasis and chemoresistance, suggest that Compound **1** can disrupt key signaling pathways important for tumor progression and dissemination.

## 5. Materials and Methods

### 5.1. Ovarian Cancer Cell Culture

Ovarian cancer cell lines SK-OV-3 (ATCC HTB-77) and OVCAR-3 (ATCC HTB-161) were purchased from ATCC^®^ and were grown in DMEM (Dulbecco’s Modified Eagle Medium, Sigma-Aldrich, Steinheim, Germany) with the addition of 10% fetal bovine serum (FBS, EuroClone, Milan, Italy) and 1% antibiotics (penicillin/streptomycin, Sigma-Aldrich, Steinheim, Germany) in an incubator at 37 °C and 5% CO_2_.

### 5.2. Compound: 3-Amino-N-(3–chloro–2–methylphenyl)–5–oxo-5,6,7,8–tetrahydrothieno[2,3-b] Quinoline–2–carboxamide

The compound was newly synthesized in the laboratory of Professor David Barker and Dr. Lisa I. Pilkington, School of Chemical Sciences, The University of Auckland, Auckland, New Zeeland. We tested four different thieno[2,3-*b*]pyridine derivatives.

### 5.3. Cytotoxicity

We performed the MTT (3-(4,5-dimethylthiazol-2-yl)-2,5-diphenyl tetrazolium bromide) test to measure the percentage of metabolically active ovarian cancer cells and to determine the half-maximal inhibitory concentration (IC_50_). The absorbances of treated cells obtained by the MTT assay were divided by the absorbances for untreated cells to obtain the percentage of metabolically active cells. Triplicates of an identical number of cells were placed on 96-well microtiter plates, and the plates were incubated for an entire night. The cells were incubated for 4, 24, 48, and 72 h after being treated with the medium (control cells) and solutions of 7 different concentrations of each derivative—0.05 µM, 0.2 µM, 0.5 µM, 1 µM, 2.5 µM, 5 µM, and 10 µM.

Following this treatment, the cells were cultured for 2 h with 0.5 mg/mL MTT solution. A microtiter plate reader (HiPo MPP-96, Biosan, Riga, Latvia) was used to measure the absorbance at 570 nm after the solution had been removed and DMSO (dimethyl sulfoxide) had been added [16]. IC_50_ was calculated using the GraphPad Prism 7.0 program (San Diego, CA, USA). Due to it having the highest efficiency, i.e., the lowest IC_50_ value, we chose to focus on 3-amino-*N*-(3-chloro-2-metilphenil)-5-oxo-5,6,7,8-tetrahydrothieno[2,3-*b*]quinoline-2-carboxamide (referred to as Compound **1** elsewhere in the text).

### 5.4. Flow Cytometry

Following the MTT tests, a concentration of 5 µM Compound **1** was applied to SK-OV-3 and OVCAR-3 cell lines for the analysis of the apoptosis and glycophenotype of CSCs and non-CSCs. This helped us obtain the IC_50_ values for 48 h for both cell lines.

#### 5.4.1. Apoptosis

An equal number of cells (1 × 10^5^ cells) were seeded in triplicate on microtiter plates with 6 wells and treated with a 5 µM concentration of the Compound **1** solution (treated cells) or with the complete medium (control cells) to perform the apoptosis test. Following the Compound **1** treatment, the cells underwent trypsinization, followed by washing in phosphate-buffered solution (PBS) and resuspension in 100 µL of binding buffer that contained 5 µL of either propidium iodide (PI) or Annexin-V-fluorescein isothiocyanate (FITC) dye (Annexin-V-FITC Apoptosis Detection Kit I, BD Biosciences, Franklin Lakes, New Jersey, USA). This type of specific cell surface staining uses Annexin-V, a protein dependent on Ca^2+^ that binds to phospholipids and serves as a marker of early apoptosis. Differentiating between early and late apoptosis was made possible by the combination of PI and Annexin-V-FITC. Based on the combination of the positive and negative outcomes of these two compounds, apoptotic, necrotic, and viable cells can be distinguished. Early apoptotic cells will have a combination of Annexin-V⁺/PI⁻, late apoptotic cells will have Annexin-V⁺/PI⁺, and viable cells will have cells that are Annexin-V^−^/PI^−^. The cells were examined using a flow cytometer (BD Accuri C6, BD Biosciences, Franklin Lakes, New Jersey, USA) after being incubated for 15 min at room temperature in the dark. FlowLogic (Inivai Victoria, Australia) was used to examine the proportion of apoptotic cells and the standard deviations.

#### 5.4.2. Determination of Glycosphingolipid Expression on Ovarian CSCs and Non-CSCs

An equal number of cells (1 × 10^5^ cells) of both cell lines, SK-OV-3 and OVCAR-3, were seeded in triplicate on 6-well microtiter plates and treated with 5 µM Compound **1** (treated cells) or complete medium (control cells) for 48 h, after which the cells of both lines were trypsinized and washed with PBS and then stained with anti-GSL antibodies. The antibodies to glycosphingolipids (GD3, nLc_4_Cer, Gg_3_Cer, Gb_4_Cer, IV^3^Neu5Ac-nLc_4_Cer, GM2, GM3, and GalNAc-GM1b) that we used were chicken polyclonal antibodies produced in the laboratory of Dr. J. Müthing [36]. The binding of primary anti-GSL antibodies was determined using secondary antibodies conjugated with eFluor 660 fluorochrome (Abcam). The fluorescence of the stained samples was measured with a flow cytometer BD Accuri C6. Data were analyzed using the FlowLogic program v. 8.0. CSCs were determined on SK-OV-3 and OVCAR-3 as CD 49f^+^. The non-CSCs cell lines were defined as CD49f^−^ on both cell lines. The expression of GSLs was determined on the same cell lines—the percentage of cells positive for the above-stated GSLs and the geometric mean value of fluorescence intensity (GMI). By using GMI on these images, we quantified the average brightness of the labeled GSLs on the cell surface. This information showed us the distribution and expression of GSLs on the cell surface, as well as how these lipids change in response to different treatments, in our case, with Compound **1**.

### 5.5. Sample Extraction, Derivatization, and Gas Chromatography–Mass Spectrometry (GC-MS)

Cells seeded in 6-well plates were treated with 5 µM Compound **1** (treated cells) or complete DMEM medium (control) for 48 h. After the incubation period, the cell medium was discarded, and the cells were thoroughly washed with buffer solution phosphate-buffered saline (PBS), followed by fixation in cold methanol, which effectively inhibited the metabolism of the cells. Without scraping the cells, cell supernatant was collected, and 20 µL of ribitol (Sigma Aldrich, Steinheim, Germany) was added as an internal standard. Finally, the samples were dried under nitrogen blowdown [37].

The derivatization process involved the addition of 25 μL of a solution consisting of 20 mg/mL metoxylamine hydrochloride in pyridine, followed by constant shaking for 60 min at 50 °C and the addition of MSTFA+1%TMCS with incubation at 50 °C for 30 min for complete derivatization. The sample was dissolved in 100 μL pyridine.

The samples were analyzed using an Agilent 8890 GC system coupled with a triple quad spectrometer system MS 7000D GC/TQ. The column we used was HP-5 MS (30 m × 0.25 mm x 0.25 μm, Agilent), with an oven program that started at 60 °C, maintained for 2 min, then increased to 210 °C at a rate of 10 °C/min, before being ramped to 240 °C at a rate of 5 °C/min, ramped to 315 °C at a rate of 25 °C/min, and, finally, held at 315 °C for 3 min.

#### GC-MS Data Preprocessing and Statistical Analysis

Agilent Mass Hunter Qualitative Analysis software v. 10.0 was used for spectral processing (including peak picking, alignment, annotation, and integration). Metabolites were identified using the NIST library. The intensity value for each metabolite was normalized to the ribitol internal standard signal.

Only compounds listed in the Human Metabolome Database (HMDB4.0) were selected. In the cell culture samples, 24 metabolites in total were found.

These two cell lines were utilized to create a panel of metabolites that were differentially expressed using MetaboAnalyst v. 6.0, a platform for metabolomics data analysis.

Student’s *t*-tests were used to determine statistical significance. A one-way ANOVA was employed to examine how Compound **1** affected the two cell lines. An unsupervised clustering technique, principal component analysis (PCA), was employed to understand overall differences in the metabolic profiles. An analysis of metabolic set enrichment (MSEA) was used to understand the connection between metabolite concentration variations and metabolic fingerprints.

### 5.6. Statistical Analysis

All data were processed by an ANOVA and a post hoc Tukey test or Kruskal–Wallis test in GraphPad Prism 7.0 (San Diego, CA, USA). Statistical significance was set at *p* < 0.05 and lower.

## Figures and Tables

**Figure 1 ijms-25-06954-f001:**
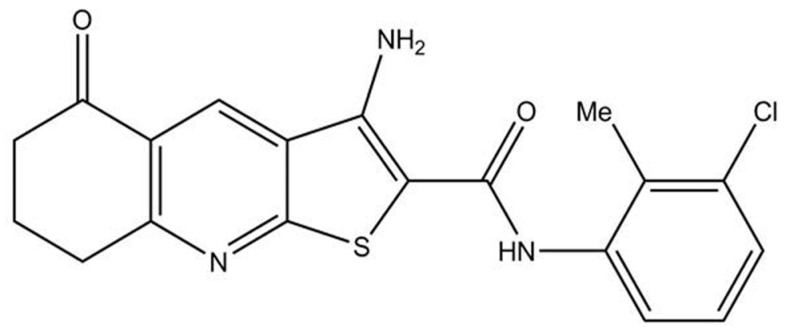
Structure of 3-amino-*N*-(3-chloro-2-methylphenyl)-5-oxo-5,6,7,8-tetrahydrothieno[2,3-*b*]quinoline-2-carboxamide (Compound **1**).

**Figure 2 ijms-25-06954-f002:**
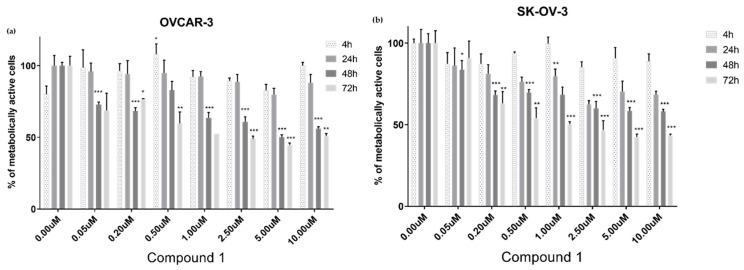
Viability of cells after exposure to Compound **1**. The OVCAR-3 (**a**) and SK-OV-3 cell lines (**b**) were treated with varying doses of Compound **1** at various times; the 3-(4,5-dimethylthiazolyl-2)-2,5-diphenyltetrazolium bromide (MTT) assay was used to measure cells’ metabolic activity. Data are shown as means from the experiment, which was performed in triplicate, ±SD. Columns, mean of viable cells; bars, SD (standard deviation); *, *p* < 0.05; **, *p* < 0.01; ***, *p* < 0.001. SD, standard deviation.

**Figure 3 ijms-25-06954-f003:**
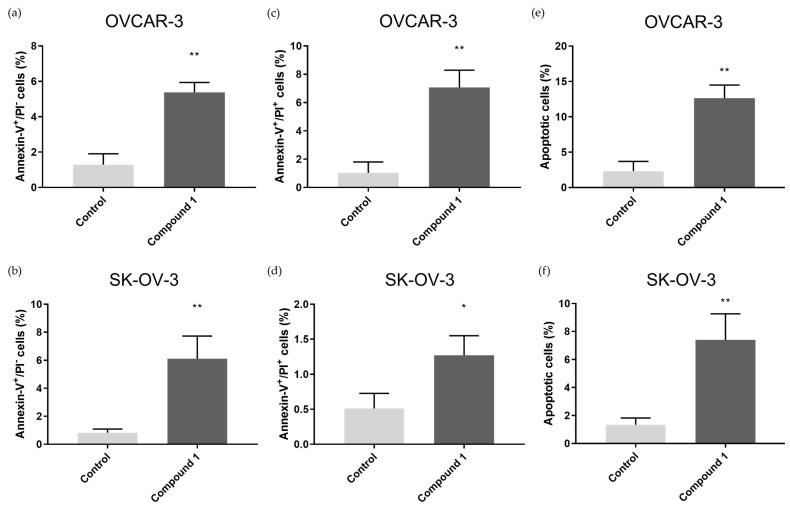
Percentage of OVCAR-3 and SK-OV-3 cell lines in early, late, and overall apoptosis. Notes: Percentage of cells in early (**a**), late (**c**), and overall (**e**) apoptosis of OVCAR-3 and early (**b**), late (**d**), and overall (**f**) apoptosis of SK-OV-3 cell lines. The displayed data are given as means from the experiment, which was performed in triplicate, ±SD. Columns, mean of cells; bars, SD; *, *p* < 0.05; **, *p* < 0.01. SD, standard deviation.

**Figure 4 ijms-25-06954-f004:**
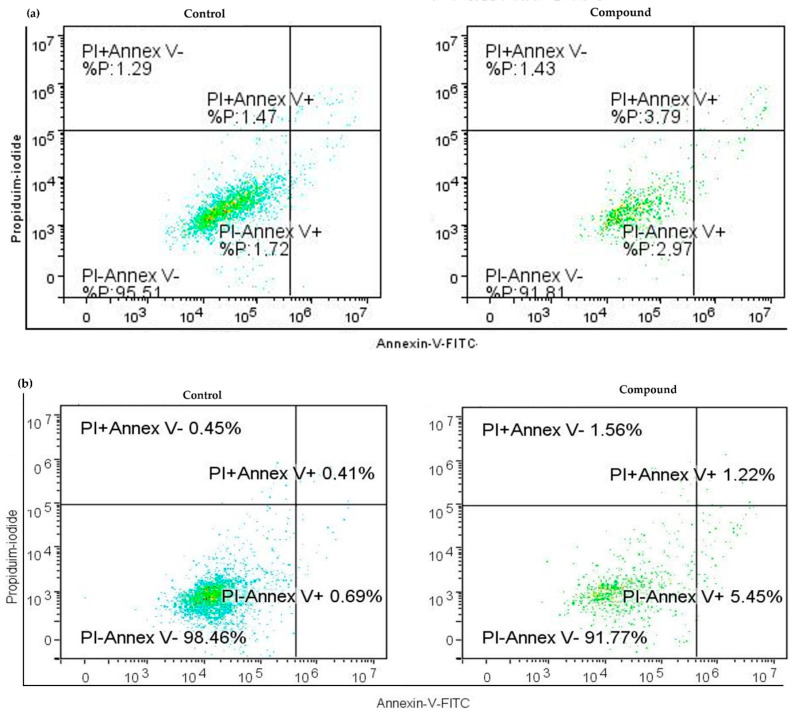
Apoptosis without (left) and with (right) Compound **1** treatment. Notes: Dot graphs of apoptotic cells in both OVCAR-3 (**a**) and SK-OV-3 (**b**) cell lines before and following exposure for 48 h. Annexin-V is shown by the *x*-axis, and propidium iodide (PI) is represented by the *y*-axis.

**Figure 5 ijms-25-06954-f005:**
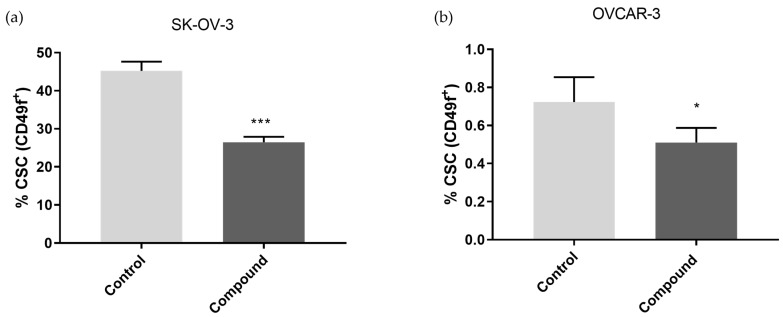
Shift of CSCs after exposure to Compound **1**. Notes: Percentage of CD49f^+^ CSCs of SK-OV-3 (**a**) and the OVCAR-3 (**b**) cell line after treatment with Compound **1** over a duration of 48 h. The displayed data are given as means from the experiment, which was performed in triplicate, ±SD. Columns, means of cells; bars, SD; *, *p* < 0.05; ***, *p* < 0.001. CSCs, cancer stem cells; SD, standard deviation.

**Figure 6 ijms-25-06954-f006:**
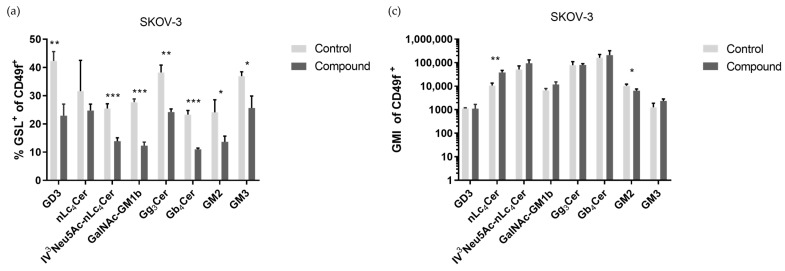
Percentage and geometric mean fluorescence intensity of glycosphingolipid positive cell subpopulations in the SK-OV-3 cell line. Notes: Percentage of CSCs (**a**) and non-CSCs (**b**) after treatment with Compound **1** over a duration of 48 h. Geometric mean fluorescence intensity of CSCs (**c**) and non-CSCs (**d**) after treatment with Compound **1** over a duration of 48 h. Data are expressed as means from the experiment, which was performed in triplicate, ±SD. Columns, means of viable cells; bars, SD; *, *p* < 0.05; **, *p* < 0.01; ***, *p* < 0.001. CSCs, CD49f^+^; non-CSCs, CD49^−^; Neu5Ac, N-acetylneuraminic acid. The gangliosides’ classification follows the IUPAC-IUB recommendations [17] and the nomenclature of Svennerholm [9]. GD3, II^3^(Neu5Ac)2-LacCer; neolactotetraosylceramide or nLc_4_Cer, IV^3^Neu5Ac-nLc_4_Cer; GalNAc-GM1b, IV^3^Neu5Ac-Gg_5_Cer; gangliotriaosylceramide or Gg_3_Cer, GalNAcβ1-4Galβ1-4Glcβ1-1Cer; globotetraosylceramide or Gb_4_Cer, GalNAcβ1-3Galα1-4Galβ1-4Glcβ1-1Cer; GM2, II^3^Neu5AcGg_3_Cer; GM3, II^3^Neu5Ac-LacCer; GMI, geometric mean fluorescence intensity; SD, standard deviation.

**Figure 7 ijms-25-06954-f007:**
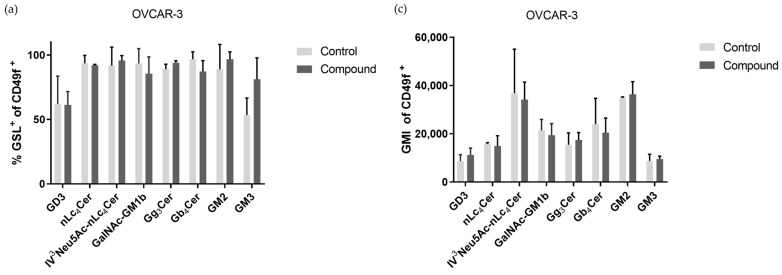
Percentage and geometric mean fluorescence intensity of glycosphingolipid-positive cell subpopulations in the OVCAR-3 cell line. Notes: Percentage of CSCs (**a**) and non-CSCs (**b**) after treatment with Compound **1** (48 h). Geometric mean fluorescence intensity of CSCs (**c**) and non-CSCs (**d**) after treatment with Compound **1** over a duration of 48 h. Data are expressed as means from the experiment, which was performed in triplicate, ±SD. Columns, means of viable cells; bars, SD; *, *p* < 0.05; **, *p* < 0.01; ***, *p* < 0.001. CSCs, CD49f^+^; non-CSCs, CD49^−^; Neu5Ac, N-acetylneuraminic acid. The designation of the gangliosides follows the IUPAC-IUB recommendations [17] and the nomenclature of Svennerholm [9]. GD3, II^3^(Neu5Ac)2-LacCer; neolactotetraosylceramide or nLc_4_Cer, IV^3^Neu5Ac-nLc_4_Cer; GalNAc-GM1b, IV^3^Neu5Ac-Gg5Cer; gangliotriaosylceramide or Gg_3_Cer, GalNAcβ1-4Galβ1-4Glcβ1-1Cer; globotetraosylceramide or Gb_4_Cer, GalNAcβ1-3Galα1-4Galβ1-4Glcβ1-1Cer; GM2, II^3^Neu5AcGg_3_Cer; GM3, II^3^Neu5Ac-LacCer; GMI, geometric mean fluorescence intensity; SD, standard deviation.

**Figure 8 ijms-25-06954-f008:**
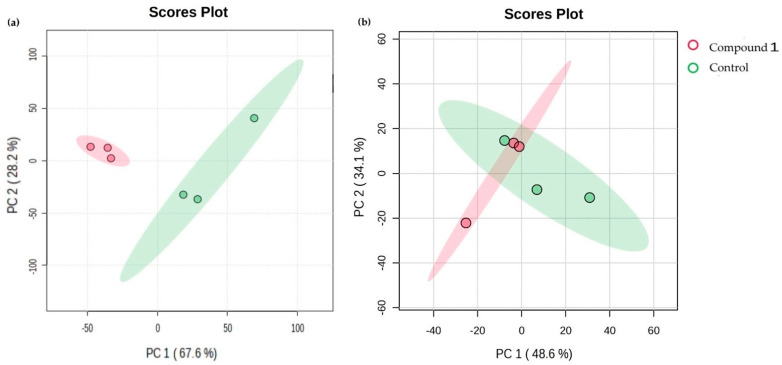
Clusters of metabolites in both treated cancer cell lines. Note: principal component analysis (PCA) of metabolic profiles of OVCAR-3 (**a**) and SK-OV-3 (**b**) cancer cell lines with or without treatment with Compound **1** over a duration of 48 h.

**Figure 9 ijms-25-06954-f009:**
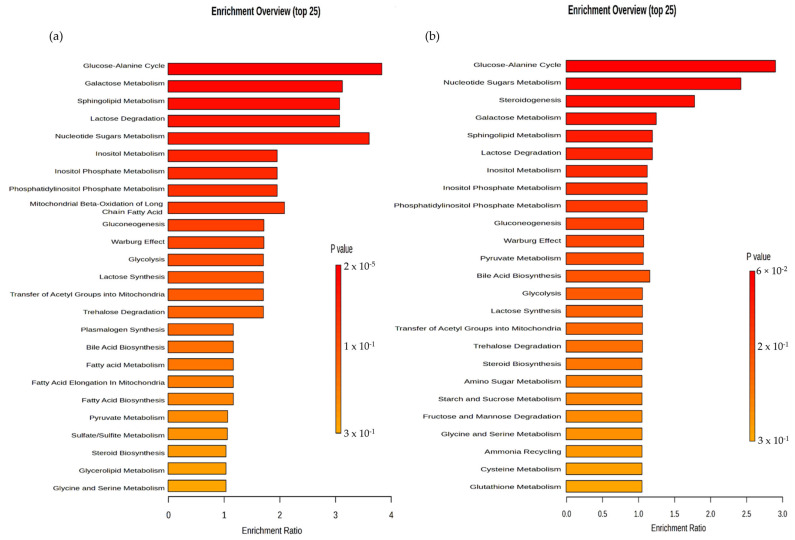
Metabolites and metabolomic pathways of OVCAR-3 and SK-OV-3 cell lines after treatment with Compound **1**. Note: metabolomic enrichment analysis of (**a**) OVCAR-3 and (**b**) SK-OV-3 cell lines after treatment with Compound **1** for 48 h.

**Figure 10 ijms-25-06954-f010:**
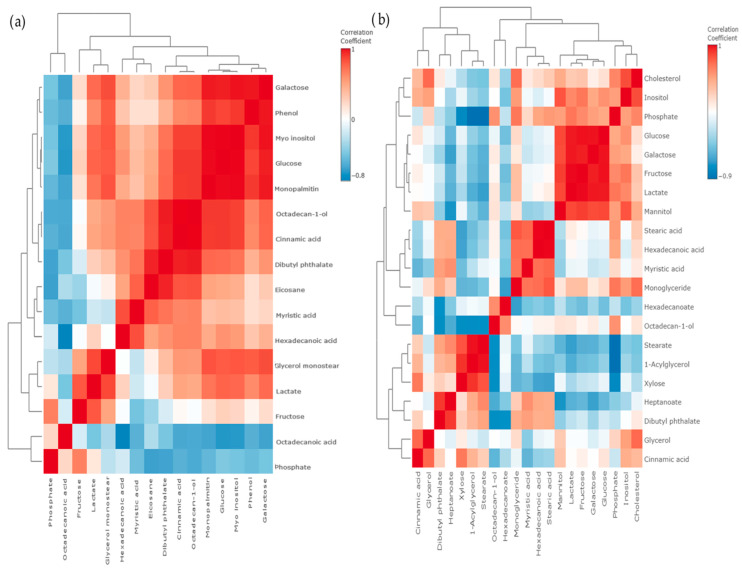
Metabolomic correlation matrix of OVCAR-3 and SK-OV-3 cell lines after treatment with Compound **1**. Notes: correlation matrix of (**a**) OVCAR-3 and (**b**) SK-OV-3 cell line metabolomics after treatment with Compound **1** for 48 h. The highest positive correlation is represented with dark red shades, and the highest negative is represented with dark blue shades. “Clusters” of substances are connected outside of the matrix.

**Figure 11 ijms-25-06954-f011:**
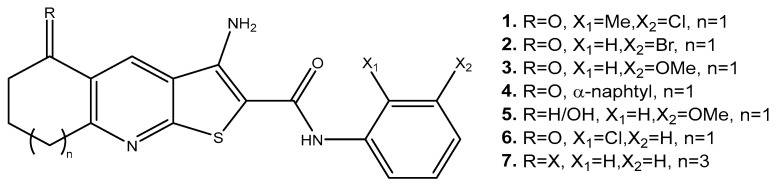
Structures of thieno[2,3-*b*]pyridine derivatives **1–7.**

**Table 1 ijms-25-06954-t001:** List of detected metabolites in OVCAR-3 and SK-OV-3 cancer cell lines after treatment with Compound **1**.

No.	Metabolite	OVCAR-3		SK-OV-3	
		*p*-Value	Fold Change	*p*-Value	Fold Change
1	Lactate	0.09	−1.13	0.06	1.43
2	Cinnamic acid	0.01 *	−4.02	0.37	5.59
3	Phenol	0.02 *	−3.12	-	-
4	Galactose	0.001 *	−1.59	0.09	0.80
5	Glucose	<0.001 *	−1.80	0.05 *	0.52
6	Dibutyl phthalate	0.09	−3.41	0.80	−0.16
7	Inositol	<0.001 *	−2.8	0.005 *	1.57
8	Octadecan-1-ol	0.01 *	−2.69	0.99	−0.01
9	Octadecanoic acid	0.09	2.41	-	-
10	Isopropyl myristate	0.003 *	−2.42	-	-
11	Heptanoate	< 0.001 *	−1.45	0.29	−0.41
12	Monoglyceride	0.05 *	−0.56	0.50	0.19
13	Eicosane	0.32	−2.98	-	-
14	Hexadecanoic acid	0.25	−1.19	0.57	−0.98
15	Phosphate	0.29	0.80	0.33	0.67
16	Fructose	0.68	−0.59	0.37	5.59
17	Stearic acid	-	-	0.71	−0.49
18	Myristic acid	-	-	0.70	−0.16
19	Stearate	-	-	0.40	−0.80
20	Mannitol	-	-	0.003 *	2.55
21	Cholesterol	-	-	0.18	0.85
22	1-Acylglycerol	-	-	0.56	−0.68
23	Glycerol	-	-	0.46	0.61
24	Xylose	-	-	0.93	0.06

Note: Following a 48 h treatment with Compound **1**, the intensity values for each metabolite in the OVCAR-3 and SK-OV-3 cells were normalized to the ribitol internal standard signal. The fold change is the ratio of treated cells’ mean signal intensity (from three separate studies) to untreated cells—the *p*-values obtained via Student’s *t*-test. * statistically significant changes.

## Data Availability

The data sets used and/or analyzed during the current study are available from the corresponding author upon reasonable request.

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
