# Peer review of "Deciphering the Interplay: Thieno[2,3-b]pyridine’s Impact on Glycosphingolipid Expression, Cytotoxicity, Apoptosis, and Metabolomics in Ovarian Tumor Cell Lines"

_ijms, 2024, doi:10.3390/ijms25136954_

Round 1

Reviewer 1 Report

Comments and Suggestions for Authors

This is a very well written research paper regarding the evaluation at the molecular and metabolomic levels of the anticancer effects of a promising  thieno-pyridine derivative. The compound confirms its anticancer activity also against stem cells.

I suggest only to:

-add data about this compound's activity in normal epithelial ovarian cells;

-Clarify the meaning of the sentence on p.14 lines 424-425 "[...] including changes in glucose metabolism and inositol levels, suggestive of pharma- cokinetics factors that affect cancer cell metabolism at multiple levels." What pharmacokinetics factors are intended?

Author Response

Dear Assistant Editor Ms. Winifred Wang and Reviewers, 

Hereby, we are submitting our revised manuscript “Deciphering the Interplay: Thieno[2,3-b]pyridine Impact on Glycosphingolipid Expression, Cytotoxicity, Apoptosis, and Metabolomics in Ovarian Tumor Cell Lines”.

We are grateful for all the reviewer comments, which helped us greatly improve our manuscript.

Here are the point-by-point answers to the reviewers’ comments:

Reviewer 1.

This is a very well written research paper regarding the evaluation at the molecular and metabolomic levels of the anticancer effects of a promising thieno-pyridine derivative. The compound confirms its anticancer activity also against stem cells.

Thank you for your positive comments. Below are point-by-point answers.

I suggest only to:

-add data about this compound's activity in normal epithelial ovarian cells;

Regarding this suggestion, unfortunately, we didn’t have the opportunity to have any normal epithelial ovarian cell line, and therefore, we are not sure about the Compound’s activity in these cells. We are aware that these data would be precious in anticipation of potential side effects of this compound in therapeutic use in the future, but we still believe that our research gives a very important insight into the mechanism of Compound’s effect in two different ovarian cancer cell lines. However, before future studies of the Compound’s efficiency in in vivo models, we will definitely include experiments on normal ovarian cells.

-Clarify the meaning of the sentence on p.14 lines 424-425 "[...] including changes in glucose metabolism and inositol levels, suggestive of pharma- cokinetics factors that affect cancer cell metabolism at multiple levels." What pharmacokinetics factors are intended?

The term "pharmacokinetics factors" in the sentence refers to various aspects related to the absorption, distribution, metabolism, and excretion (ADME) of Compound 1 within cancer cells. Specifically, we hypothesize that the observed changes in glucose metabolism and inositol levels following treatment with Compound 1 are indicative of its influence on the pharmacokinetic properties of cancer cells. These alterations suggest that Compound 1 may impact cancer cell metabolism at multiple levels through its pharmacokinetic characteristics.

Reviewer 2 Report

Comments and Suggestions for Authors

The article presents interesting results from the study of the impact of anticancer compounds on ovarian tumor cell lines. The scientific value of the article  at the required IJMS journal level. 

However, the manuscript needs to be completely rewritten to the academic English style. The construction of sentences did not respond to the Englisch language, especially in the parts introduction, and results, (chapter 2.1- 2.4). 

Further suggested correction:

In Table 1. the expression Mystric acid is an unusual chemical  name, it is  correct?

Figure 9 and 10 are hard to read, it shoul be in bigger size.

Comments on the Quality of English Language

The manuscript needs to be completely rewritten to the academic English style. We recommend that authors have their manuscripts checked by an English language native speaker before final approval of their submission;

Author Response

Dear Assistant Editor Ms. Winifred Wang and Reviewers,

Hereby, we are submitting our revised manuscript “Deciphering the Interplay: Thieno[2,3-b]pyridine Impact on Glycosphingolipid Expression, Cytotoxicity, Apoptosis, and Metabolomics in Ovarian Tumor Cell Lines”.

We are grateful for all the reviewer comments, which helped us greatly improve our manuscript.

Here are the point-by-point answers to the reviewers’ comments:

Reviewer 2.

The article presents interesting results from the study of the impact of anticancer compounds on ovarian tumor cell lines. The scientific value of the article at the required IJMS journal level. 

Thank you for your positive comments. Below are point-by-point answers.

-However, the manuscript needs to be completely rewritten to the academic English style. The construction of sentences did not respond to the English language, especially in the parts introduction, and results, (chapter 2.1- 2.4). 

We are very surprised by your comment on the quality of the English language, since two co-authors, academic professors (Lisa Pilkington and David Barker) who are native speakers went through the whole manuscript and edited it according to the recommended style. The other reviewer also commented that the manuscript is very well written. However, prof. Barker revised the manuscript again and we are now submitting this version with highlighted sentences that were edited. I hope you find this version improved according to the language style.

Further suggested correction:

-In Table 1. the expression Mystric acid is an unusual chemical name, it is correct?

Thank you for the comment: the expression “Mystric” was changed to “Myristic” in Table 1.

-Figures 9 and 10 are hard to read, it should be in bigger size.

The suggestion has been accepted and the letter font has been increased.

Comments on the Quality of English Language

The manuscript needs to be completely rewritten to the academic English style. We recommend that authors have their manuscripts checked by an English language native speaker before final approval of their submission;

Prof. David Barker, who is an English language native speaker, checked and revised the manuscript and we are now submitting this version of the manuscript.

Round 2

Reviewer 1 Report

Comments and Suggestions for Authors

The authors did not improve the manuscript according to my suggestions. If they were not able to evaluate cellular selectivity on epithelial ovarian cells as suggested, they could have used another normal cell line such as fibroblasts,  to have indications to address the issue. Moreover, they did not changed the  last conclusions sentence. The pharmacokinetics properties indoubtly are the drug ADME properties, but the fact that the molecule can modify some cellular pathways is definitevely not a prediction of its pharmacokinetics. Therefore, that sentence should be removed or adjusted accordingly.

Author Response

Reviewer 1

The authors did not improve the manuscript according to my suggestions. If they were not able to evaluate cellular selectivity on epithelial ovarian cells as suggested, they could have used another normal cell line such as fibroblasts,  to have indications to address the issue. Moreover, they did not changed the  last conclusions sentence. The pharmacokinetics properties indoubtly are the drug ADME properties, but the fact that the molecule can modify some cellular pathways is definitevely not a prediction of its pharmacokinetics. Therefore, that sentence should be removed or adjusted accordingly.

  • Thank you, a section “Toxicological profile” has been added to the manuscript. A panel of the thieno[2,3-b] pyridines was used with the MCF-10A non-cancer mammary epithelial cell line and the MDA-MB-435 melanoma cell line was used for comparison. Interpretation of the data is given and the selectivity of the thieno[2,3-b] pyridines for cancer cells is discussed. Furthermore, in vivo mouse toxicity data in the supplementary file from the NCI was added for two thieno[2,3-b] pyridine derivatives.
  • The last paragraph in the conclusion has now been removed as suggested by the reviewer.

Reviewer 2 Report

Comments and Suggestions for Authors

 I accept your revision of article

Author Response

Thank you for your positive answers and the time you took to review our manuscript

Round 3

Reviewer 1 Report

Comments and Suggestions for Authors

Some efforts were done, especially regarding the toxicity evaluation. It is not clear if these studies were previously performed-published. If so, please cite the relevant literature and briefly add comments in the discussion. Regarding the evaluation on MCF-10A cells, it is not useful, it is a completely different cellular model with respect to the ovarian tumour cell lines. It can be avoided, considering the inclusion of a toxicity evaluation.

Author Response

REVIEWER 1

Some efforts were done, especially regarding the toxicity evaluation. It is not clear if these studies were previously performed-published. If so, please cite the relevant literature and briefly add comments in the discussion. Regarding the evaluation on MCF-10A cells, it is not useful, it is a completely different cellular model with respect to the ovarian tumour cell lines. It can be avoided, considering the inclusion of a toxicity evaluation.

Dear reviewer,

We appreciate your comment regarding the use of MCF-10A cells. Indeed, MCF-10A cells are not a perfect representation of ovarian tissue. However, while MCF-10A cells are not ovarian, they provide a valuable comparison point as a normal, non-transformed epithelial cell line. These data were not published previously. Nevertheless, we agreed with your comment and have removed the data from the MCF-10A cells included in version 2 of the manuscript.

We also added reference 19 (Arabshahi, H.J.; Leung, E.; Barker, D.; Reynisson, J. The Development of Thieno[2,3-b]Pyridine Analogues as Anticancer Agents Applying in Silico Methods. MedChemComm 2014, 5, 186–191, doi:10.1039/C3MD00320E.) including previously reported results of selective toxicity of Compound 1 in various cell lines, as a support of its toxicological profile.

We briefly commented on these results in the discussion chapter, as suggested.